

# Impacts of experimental conditions on soil saturated hydraulic
# conductivity in conventional and conservation tillage practices
Kaihua Liao[1,2*], Xiaoming Lai[1,2], Qing Zhu[1,2,3*]
[1]Key Laboratory of Watershed Geographic Sciences, Nanjing Institute of Geography
and Limnology, Chinese Academy of Sciences, Nanjing 210008, China
[2]University of Chinese Academy of Sciences, Beijing 100049, China
[3]Jiangsu Collaborative Innovation Center of Regional Modern Agriculture &
Environmental Protection, Huaiyin Normal University, Huaian 223001, China

[*] Corresponding author. Tel.: +86 25 86882139; fax: +86 25 57714759.
*E-mail addresses:* khliao@niglas.ac.cn (K. Liao); qzhu@niglas.ac.cn (Q. Zhu)



**Abstract.** The saturated hydraulic conductivity ($K_{sat}$) is a key soil hydraulic property
governing agricultural production. However, the influence of conversion from
conventional tillage (CT) to conservation tillage (CS) (including no tillage (NT) and
reduced tillage (RT)) on $K_{sat}$ of soils is not well understood and still debated. In this
study, we applied a global meta-analysis method to synthesize 201 paired
observations for soil $K_{sat}$ from 59 published studies, and investigated factors
influencing the effects of conversion to CS on $K_{sat}$. Results showed that the $K_{sat}$
measured by hood infiltrometer, tension disc infiltrometer, and Guelph permeameter
produced a similar pattern under CS practices, with non-significant ($p > 0.05$)
increase of 6.6%, 3.6% and 4.9%, respectively. However, conversion to CS
significantly ($p < 0.05$) increased $K_{sat}$ by 32.0% for ring infiltrometer, while it
decreased $K_{sat}$ by 3.2% for constant/falling head ($p > 0.05$). Soil layer, CS type and
soil texture had no significant ($p > 0.05$) effects on the influence of conversion to CS
on the $K_{sat}$, but the $K_{sat}$ under CS showed a greater increase for a longer conversion
period (time since conversion). In addition, mean annual temperature (MAT) was
found to be an important driver controlling the response of $K_{sat}$ to tillage conversion at
the large scale. These findings suggested that quantifying the effects of tillage
conversion on soil $K_{sat}$ needed to consider experimental conditions, especially the
measurement technique and conversion period.



## 1 Introduction

The saturated hydraulic conductivity ($K_{sat}$), which reflects soil permeability when the soil is saturated, is critical for calculating water flux in soil profile and designing irrigation and drainage systems (Bormann and Klaassen, 2008). It is also an essential soil parameter in agro-ecological, hydrological and biogeochemical models across different scales. The $K_{sat}$ changes greatly in space and time due to factors such as texture, organic matter content, bulk density, porosity, vegetation types or tillage practices (Schaap et al., 1998; Zhu et al., 2014; Liao et al., 2018; Schlüter et al., 2020). Infiltration experiments are often applied to measure infiltration rate of soils in field by different techniques, such as hood infiltrometer (Schwärzel and Punzel, 2007), tension disc infiltrometer (Perroux and White, 1988) and single- or double-ring infiltrometer (Bouwer, 1986). In addition, permeameters are also adopted to measure $K_{sat}$, such as Guelph permeameter (Reynolds and Elrick, 1985) used in field and constant/falling head permeameter applied on intact (undisturbed) or repacked soil cores (Klute and Dirksen, 1986).

Tillage is one of the main causes of spatio-temporal variability in $K_{sat}$. Conventional tillage (CT), mainly refers to as heavy tillage practices down to 25–30 cm soil depths, is a widely adopted management practice which could significantly affect soil aggregation and hydraulic properties (Pittelkow et al., 2014; Li et al., 2019). Conservation tillage (CS) is often defined as no-tillage (NT) or reduced tillage (RT) with/without residue retention. NT is confined to soil disturbance associated with crop seeding or planting, while in RT a cultivator or disc harrow is used to loosen the soil



superficially (Licht and Al-Kaisi, 2005). The CS practices directly affect soil physical
properties by increasing residue retention and decreasing soil disturbance (Turmel et
al., 2015). The conversion from CT to CS has been demonstrated to improve physical
environment of the soil (Li et al., 2019). In a wheat/soybean–corn rotation field in the
Argentinian Pampas, Sasal et al. (2006) found that aggregates of silty cultivated soils
were 30% more stable in CS than under CT due to 21% increase in organic matter.
Based on long-term wheat-fallow tillage experiments, Blanco-Canqui et al. (2009)
observed that the near-surface soil maximum bulk density of the CT was higher than
that of the NT soil by about 6% at Akron, Hays, and Tribune in the central Great
Plains. However, it is still controversial whether the change from CT to CS can
increase $K_{sat}$. Several studies (Jarecki and Lal, 2005; Abid and Lal, 2009; Nouri et al.,
2018) have reported systematic improvements in the $K_{sat}$ under CS practices, which
may be attributed to the decomposition of aggregates, the formation of surface seal by
the raindrop impact, the increase of compactness and the decrease of average
pore-size distribution of topsoil under CT. In contrast, pores in CS soil may be well
connected and protected from raindrop impact and other disturbances by residual
mulch (Blanco-Canqui and Lal, 2007; Shukla et al., 2003). However, other studies
have shown that $K_{sat}$ under CS is not higher than that under CT (Anikwe and Ubochi,
2007; Abu and Abubakar, 2013; Busari, 2017). Tillage conversion may also lead to
different degrees of changes in the factors (e.g., soil structure, texture and bulk
density) influencing $K_{sat}$ (Cameira et al., 2003). There, the response of $K_{sat}$ to tillage
was complex and not well understood. In addition to CS practices, there are many





other agricultural practices that may increase $K_{sat}$, such as compost addition, straw
returning and biochar returning (Olson et al., 2013; Xiao et al., 2020). However,
addressing these agricultural practices is beyond the scope of this study.

The effects of tillage on $K_{sat}$ may partly depend on measurement techniques

(Morbidelli et al., 2017). The $K_{sat}$ measured by different measurement techniques may
differ by an order of magnitude, which is mainly due to the following reasons: (1) the
geometry of water application to the soil is different; (2) the strategies to prevent
surface sealing and pore plugging are different; (3) the soil wetted (or saturated)
volume is different; and (4) for laboratory procedures, the sample size and sampling
method may alter the soil core conditions (Fodor et al., 2011; Schlüter et al., 2020).
The uncertainty of measurement techniques can mask the influence of the conversion
from CT to CS on $K_{sat}$. Soil layer, texture and CS type may also influence the tillage
effect on $K_{sat}$ (Alletto et al., 2010). For example, Yu et al. (2015) observed that tillage
of cropland created temporarily well-structured topsoil but compacted subsoil as
indicated by low subsoil $K_{sat}$. Soil texture is one of the main factors controlling soil
infiltration and hydraulic conductivity. Coarse textured soils lose moisture much more
easily than fine textured soils because of the weaker capillary forces in the large pore
spaces. CS has direct and indirect effects on soil structure. Generally, soil compaction
begins with the conversion to CS, which may lead to a decrease in air capacity and
increase bulk density and permeability resistance of surface soil (Abdollahi and
Munkholm, 2017). However, there has not yet been a global synthetic analysis
specifically focusing on how environmental conditions could affect the tillage effect



on $K_{sat}$. Recently, Li et al. (2019) applied a global meta-analysis to investigate the
direction and magnitude of changes in $K_{sat}$ in response to CS practices. They found
that CS practices improved $K_{sat}$ in croplands compared with CT. However, the
generalizable patterns and regulating factors of tillage effects on $K_{sat}$ remain unclear at
the global scale. Therefore, it is necessary to synthesize all available data to reveal
global-scale response of $K_{sat}$ and to identify the main regulating factors for its
response under CS practices.

The objective of this study was to detect the influences of different experimental

conditions (i.e., measurement technique, soil layer, texture, CS type, conversion
period, mean annual precipitation or MAP, mean annual temperature or MAT and
elevation) on the effects of conversion from CT to CS on the $K_{sat}$ based on a global
meta-analysis of 59 studies.
**2 Materials and methods**
**2.1 Source of data and selection criteria**
Peer-reviewed journal articles and dissertations related to $K_{sat}$ under CT and CS were
searched using Web of Science and China National Knowledge Infrastructure (CNKI,
http://www.cnki.net) through June 30, 2020. The keywords used for the literature
search were related to: "saturated hydraulic conductivity", "steady-state infiltration
rate", "conventional tillage", "conservation tillage", and "till". Using these keywords,
a total of 107 papers were searched. To minimize bias, our criteria were as follows: (1)
the selected articles included paired observations comparing CT and CS based on
field experiments; (2) specific CS practices included RT and NT; (3) other agronomic



measures, such as residue retention and film mulching, must be similar between
paired controls (CT) and treatments (CS) during the selection process; (4) means,
standard deviations (SD) (or standard errors (SE)) and sample sizes were directly
provided or could be calculated from the studies; (5) if one article contained $K_{sat}$ in
multiple years, only the latest results were applied since the observations should be
independent in the meta-analysis (Hedges et al., 1999); (6) for Guelph permeameter,
only the one-head technique was considered for meta-analysis. Previous studies
(Reynolds and Elrick, 1985; Jabro and Evans, 2006) have shown that for a significant
percentage of times, the two-head method produced unreliable results when using
Guelph permeameter. In total, 59 published studies conducted around the world were
selected from 107 published articles (Fig. 1). The locations of these studies and their
site information are presented in Tables S1 and S2.

Of the 59 studies, 7 did not provide $K_{sat}$ values. These 7 studies only provided the

steady-state infiltration rate, which was assumed to be the $K_{sat}$ by convention in this
study (Yolcubal et al., 2004; Kirkham, 2014) (Table S2). A total of 5 measurement
techniques for infiltration rate and $K_{sat}$ were involved in these 59 studies, including
hood infiltrometer, tension disc infiltrometer, ring infiltrometer, Guelph permeameter
used in field, and constant/falling head applied on undisturbed soil cores. The first
three techniques determined infiltration rate based on water entry into an unsaturated
soil at the soil-atmosphere boundary, while the last two measured the flow of water
from one point to another within the soil mass. The final infiltration rate measured by
a single or double ring infiltrometer and by tension and hood infiltrometer methods at



zero tension were often equated to $K_{sat}$ of the soil. In the selected literature, the
infiltration rate has been converted to $K_{sat}$ for the first three techniques.
**2.2 Data extraction and statistical analysis**
For each study, the mean, the standard error (SE) or standard deviation (SD), and
sample size values for treatment and control groups were extracted for $K_{sat}$. The units
of $K_{sat}$ for all studies were converted to cm d$^{-1}$. For studies that did not provide SD or
SE, SD was often predicted as 0.1 times the mean in previous studies (Li et al., 2019).
Considering the relatively strong spatial variability of soil $K_{sat}$, we set the SD value as
0.4 in this study. In addition to $K_{sat}$, the measurement technique of $K_{sat}$, soil depth,
texture, CS practices, conversion period (time since the conversion), MAP, MAT and
elevation were also recorded if they could be obtained. All data were extracted from
words, tables or digitized from graphs with the software GetData v2.2.4
(http://www.getdata-graph-digitizer.com).
The METAWIN 2.1 software (Sinauer Associates Inc., Sunderland, MA, USA)
(Rosenberg et al., 2000) was used to perform meta-analysis in this study. The natural
logarithm of the response ratio ($R$) was used to estimate the effects of changes in
tillage practices on $K_{sat}$ (Hedges et al., 1999):
$$\ln(R) = \ln\left(\frac{\overline{X_s}}{\overline{X_t}}\right) = \ln\left(\overline{X_s}\right) - \ln\left(\overline{X_t}\right) \qquad (1)$$
where $\overline{X_s}$ and $\overline{X_t}$ are the mean value of $K_{sat}$ under CS (treatment) and CT practices
(control), respectively. The natural log was applied for meta-analysis since its bias is
relatively small and its sampling distribution is approximately normal (Luo et al.,
2006). In addition, the variance ($VAR$) of $\ln(R)$ was calculated as:



$$VAR = \frac{S_s{}^2}{n_s \overline{X_s}^2} + \frac{S_t{}^2}{n_t \overline{X_t}^2} \qquad\qquad (2)$$
where $n_s$ and $n_t$ are the sample sizes for the CS and CT practices, respectively; and
$S_s$ and $S_t$ are the SDs for CS and CT practices, respectively. To examine whether
experimental conditions (including measurement technique, soil layer, texture and CS
type) alter the response direction and magnitude of $K_{sat}$, observations were divided
into subgroups according to the measurement techniques (hood infiltrometer, tension
disc infiltrometer, Guelph permeameter, ring infiltrometer used in field and
constant/falling head used on undisturbed soil cores), soil layer (surface (0-20 cm)
and subsurface (> 20 cm depth)), CS practices (NT and RT), soil texture (fine-,
medium-, and coarse-textured soil) and conversion period (1-5 yr, 6-10 yr, 11-15 yr,
16-20 yr, 21-30 yr and > 30 yr). For differentiating among soil textural classes, we
applied the United States Department of Agriculture (USDA) soil textural triangle,
and considered clay, sandy clay, and silty clay soils as fine texture; silt, silt loam, silty
clay loam, loam, sandy clay loam, and clay loam soils as medium texture; and sand,
loamy sand, and sandy loam soils as coarse texture (Daryanto et al., 2016).
A random effects model with a grouping variable was used to compare responses
among different subgroups. In this model, there are two sources of variance, including
within-study variance (*VAR*) and between-study variance ($\tau^2$), both of which were
used to calculate the weighting factor $\omega$ = [1/(*VAR*+$\tau^2$)], with $\tau^2$ = (*Q-df*)/*C*, where *Q*
is the observed weighted sum of squares, *df* are the degrees of freedom, and *C* is a
normalization factor. The calculation equations of *Q*, *df* and *C* can be referred to
Borenstein et al. (2010). The weighted ln(*R*) (ln(*R*[*]), which was used as the effect




size, was then determined based on the $\omega$. $\ln(R^*)$ is defined as
$\ln(R^*) = \sum_{i=1}^{m}[\omega_i \ln(R_i)]/\sum_{i=1}^{m}\omega_i$, where $\omega_i$ and $\ln(R_i)$ are $\omega$ and $\ln(R)$ of the $i$th
observation, respectively. The $\ln(R^*)$ value indicated the magnitude of the treatment
impact. Positive or negative $\ln(R^*)$ values represented an increase or decrease effect of
the tillage treatment, respectively. Zero meant no difference between treatment (CS)
and control (CT) group. Finally, resampling tests were incorporated into our
meta-analysis using the bootstrap method (999 random replicates). The mean effect
size ($\overline{\ln(R^*)}$, calculated from 999 iterations) and 95% bootstrap confidence intervals
(CI) were generated. If the 95% CI values of $\ln(R^*)$ did not overlap zero, the effect of
changes in tillage practices on $K_{sat}$ were considered significant at $p < 0.05$. The
percentage change between CS and CT was calculated as $\exp[\overline{\ln(R^*)}]$-1.
Linear regression analyses were performed by SPSS software (version 13.0,
SPSS Inc., Chicago, Illinois, USA) to evaluate the relationships between the $\ln(R)$ for
soil saturated hydraulic conductivity under CS with MAP, MAT and elevation.
**3 Results**
The mean effect sizes of $K_{sat}$ under CS were 0.064 (95% CI: -0.519 to 0.681), 0.035
(95% CI: -0.078 to 0.144), 0.278 (95% CI: 0.084 to 0.508), 0.048 (95% CI: -0.156 to
0.253) and -0.033 (95% CI: -0.201 to 0.138) for hood infiltrometer, tension disc
infiltrometer, ring infiltrometer, Guelph permeameter and constant/falling head,
respectively (Fig. 2a). The $K_{sat}$ measured by hood infiltrometer, tension disc
infiltrometer, and Guelph permeameter showed a similar pattern, with non-significant
($p > 0.05$) increase of 6.6%, 3.6% and 4.9%, respectively. However, conversion from



CT to CS significantly ($p < 0.05$) increased $K_{sat}$ by 32.0% for ring infiltrometer used
in field, while it decreased $K_{sat}$ by 3.2% for constant/falling head ($p > 0.05$).

Surface and subsurface $K_{sat}$ showed a similar pattern under CS, with

non-significant ($p > 0.05$) increase of 6.8% and 6.1%, respectively (Fig. 2b). The
reverse response of $K_{sat}$ to both CS practices was observed. Conversion to NT
increased $K_{sat}$ by 3.4% ($p > 0.05$), whereas conversion to RT decreased $K_{sat}$ by 6.5%
($p > 0.05$) (Fig. 2c). For coarse-, medium- and fine-textured soils, changes in tillage
practices increased $K_{sat}$ by 3.0%, 1.2% and 6.4% ($p > 0.05$), respectively. In addition,
the mean effect sizes of $K_{sat}$ under CS were -0.177 (95% CI: -0.331 to -0.031), 0.144
(95% CI: 0.010 to 0.278), 0.231 (95% CI: 0.046 to 0.444), 0.096 (95% CI: -0.690 to
0.739), 0.199 (95% CI: -0.237 to 0.617) and 0.427 (95% CI: 0.036 to 0.857) for 1-5 yr,
6-10 yr, 11-15 yr, 16-20 yr, 21-30 yr and > 30 yr after conversion, respectively.

The $\ln(R)$ of $K_{sat}$ decreased significantly with MAT ($p < 0.001$; Fig. 3a), whereas

no significant correlation was found between the $\ln(R)$ of $K_{sat}$ and MAP and elevation
($p = 0.123$ and $p = 0.262$, respectively; Fig. 3bc).
**4 Discussion**
The change of $K_{sat}$ caused by the conversion from CT to CS varied between the
different measurement techniques employed (Fig. 2a). Our findings implied that the
measurement technique had an important influence on the determination of $K_{sat}$
(Reynolds et al., 2000; Rienzner and Gandolfi, 2014). Many previous studies found
that $K_{sat}$ measured with hood infiltrometer was substantially higher than that measured
with tension disc infiltrometer (Matula et al., 2015; Schlüter et al., 2020). It is because



the hood infiltrometer measurements were conducted directly on field-moist soil,
while fine particles of the contact material may cause clogging of pores when using
tension disc infiltrometer (Schwärzel and Punzel, 2007). This study found that the
increase of $K_{sat}$ measured by hood and tension disc infiltrometer methods was similar
when CT was changed to CS. Considering the CI of the effect sizes of $K_{sat}$ was wide
for the hood (Fig. 2a), more data are needed to verify this conclusion in the future.

The increase of $K_{sat}$ measured by single- or double-ring infiltrometer was

substantially larger than the other two types of infiltrometer. This is consistent with
the study by Buczko et al. (2006), who also found that the $K_{sat}$ measured with the ring
infiltrometer were higher than the corresponding values measured with the tension
infiltrometer. These differences may be caused by subcritical soil water repellency
(i.e., contact angles of the soil-water-air interface below $90^{o}$), and other factors, such
as air entrapment and differences in water saturation. Another reason could be that the
ring infiltrometer had a deeper water infiltration depth and bigger infiltration area.
Overall, conversion to CS generally increased $K_{sat}$ measured by the three
infiltrometers. This is probably due to the aggregate destruction and formation of
surface seals in the CT soil (Fodor et al., 2011). In addition, CT corresponded to lower
organic matter of the soil and aggregate stability (Azooz and Arshad, 1996).
Conversely, the $K_{sat}$ measured by constant/falling head permeameter generally
decreased under CS. The reason may be that the soil volume measured by this method
is small, and the macropore channel may be cut off during the sampling process, thus
reducing the $K_{sat}$. In addition, the constant/falling head method using intact core





samples had a strong variability, which can affect the measurement results (Soracco et
al., 2010). Moreover, the lack of vacuum when saturating the soil core sample prior to
constant/falling head measurements could cause air entrapment, which greatly reduces
the hydraulic conductivity (Faybishenko, 1995; Steenhauer et al., 2011). The above
results suggested to us that the measurement technique had an important influence on
the response of $K_{sat}$ to tillage conversion. Recently, Schlüter et al. (2020) also found
that the increase in $K_{sat}$ caused by a higher abundance of large biopores under NT was
only detected with hood infiltrometer measurements in the field and reversed in
tension disc infiltrometer measurements on undisturbed soil cores.
Our results showed that the conversion period substantially affected the $K_{sat}$. It is
noted that conversion from CT to CS significantly ($p < 0.05$) decreased $K_{sat}$ for 1-5 yr.
The possible reason is that soil compaction set in with the conversion to CS, which
can lead to a reduction in macroporosity and an increase in bulk density and
microporosity. Many previous studies have demonstrated the negative relationship
between bulk density and $K_{sat}$ (e.g., Vereecken et al., 1989; Huang et al., 2021). In this
case, initially bulk density increased, while $K_{sat}$ decreased. However, after several
years this reversed through a re-structuring of the soil by bioturbation (Schlüter et al.,
2020). In addition, the decreased soil disturbance with long-term CS practices can
improve soil organic carbon accumulation over time, which also leaded to better
water infiltration (Six et al., 2000; Li et al., 2019).
The response of $K_{sat}$ was negatively correlated with MAT, indicating that this
variable had potential controls on the $K_{sat}$ responses to tillage conversion. The

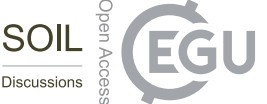

possible reason is that mean annual temperature mainly indirectly control $K_{sat}$
responses via other variables (e.g., biological processes and effective porosity). Based
on these results, we argue that in the cold and temperate regions, the improvement of
$K_{sat}$ by tillage conversion will be greater than that in the tropical regions. Although
this study provided a global meta-analysis of the responses of $K_{sat}$ to changes in tillage
practices under different experimental conditions, the magnitude of these responses
might be uncertain. For example, a relatively small number of observations were
obtained with the hood infiltrometer, which would affect the results of meta-analysis.
Nevertheless, this study emphasized the importance of experimental conditions in
judging the change of tillage practices for enhancing soil permeability.
**5 Conclusions**
Our global meta-analysis indicated that conversion from CT to CS had generally
positive effects on $K_{sat}$. However, these effects were related to experimental
conditions, especially the measurement technique, conversion period and MAT. The
increase of $K_{sat}$ measured by single- or double-ring infiltrometer was substantially
larger than the other techniques. In addition, the $K_{sat}$ under CS showed a greater
increase for a longer conversion period. Moreover, the lower the MAT, the more
obvious the improvement effect of tillage conversion on $K_{sat}$. Our findings should be
useful for understanding the underlying mechanisms driving the change of $K_{sat}$ with
CS practices.
**Data availability.** The data that support the findings of this study are available from
the corresponding author upon request.



**Author contributions.** KL designed this study, KL and XL performed the
meta-analysis, KL and QZ obtained funding, and KL wrote the paper with
contributions from QZ.
**Competing interests.** The authors declare that they have no conflict of interest.
**Acknowledgements.** We thank two anonymous reviewers and editor for their efforts
on this paper. Support for this research was provided by the National Natural Science
Foundation of China and by Chinese Academy of Sciences.
**Financial support.** This study was financially supported by the National Natural
Science Foundation of China (42125103 and 42171077), and the Youth Innovation
Promotion Association, Chinese Academy of Sciences (2020317).
**Review statement.** This paper was reviewed by two anonymous referees.

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





# List of Figures:

**Figure 1:** The geographical coverage of the 59 studies used in the meta-analysis.

**Figure 2:** Factors influencing the effect sizes of the soil saturated hydraulic conductivity under conservation tillage (CS) from a global meta-analysis of 59 studies. The error bars indicate effect sizes and 95% bootstrap confidence intervals (CI). The effect of CS was statistically significant if the 95% CI did not bracket zero. The sample size for each variable is shown next to the bar.

**Figure 3:** Relationships between the natural logarithm of the response ratio ($\ln(R)$) for soil saturated hydraulic conductivity under conservation tillage with (a) mean annual temperature (MAT), (b) mean annual precipitation (MAP) and (c) elevation.



Figure 1

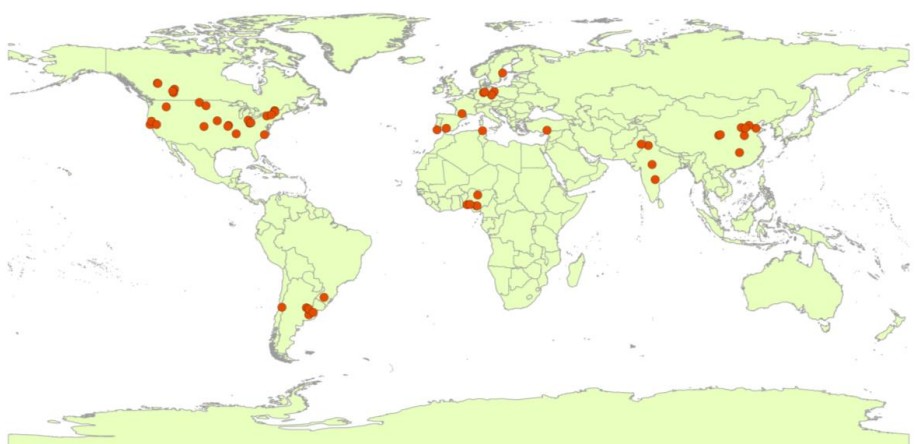

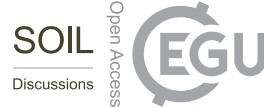

Figure 2

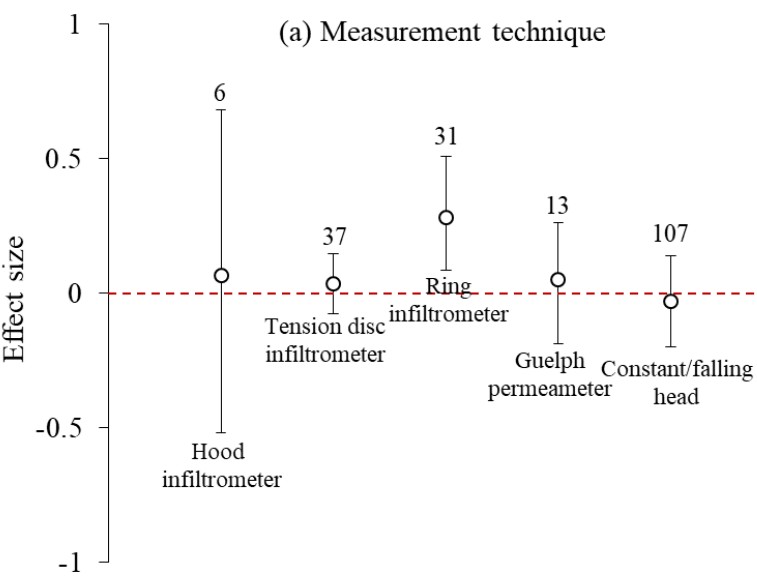

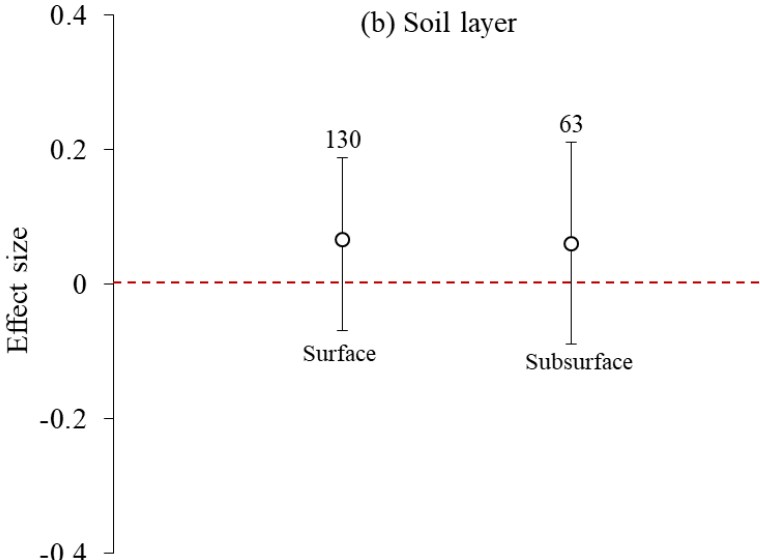



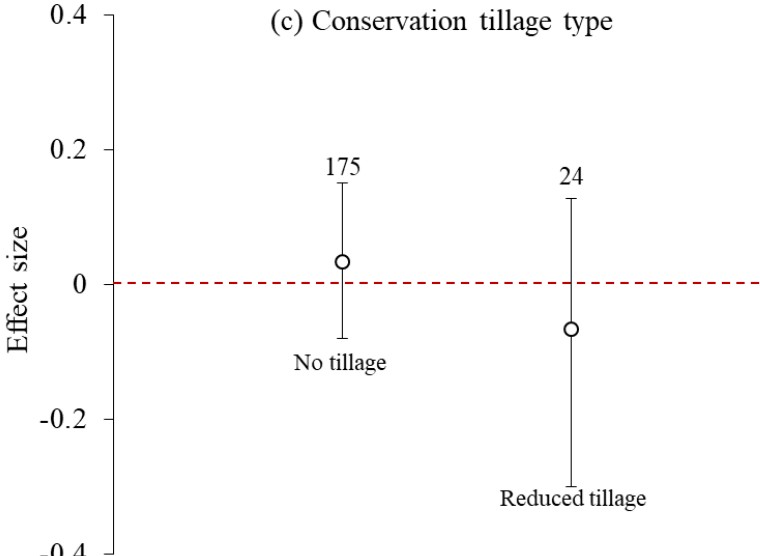

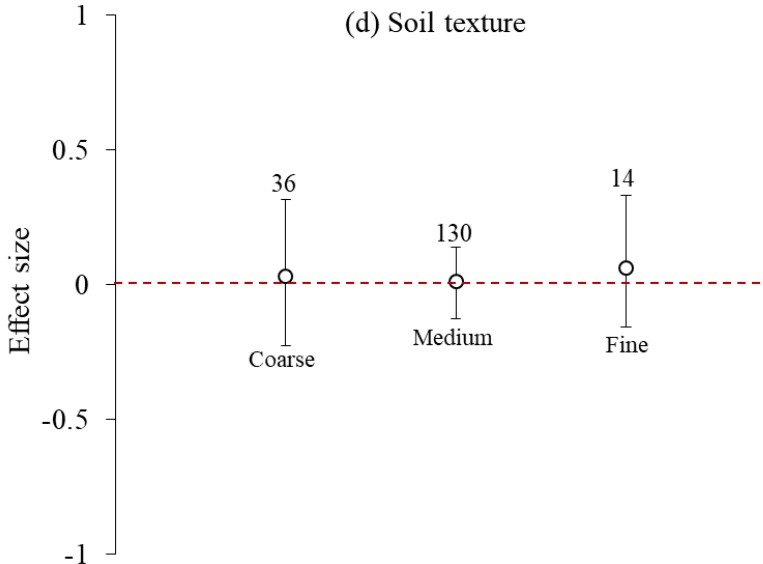

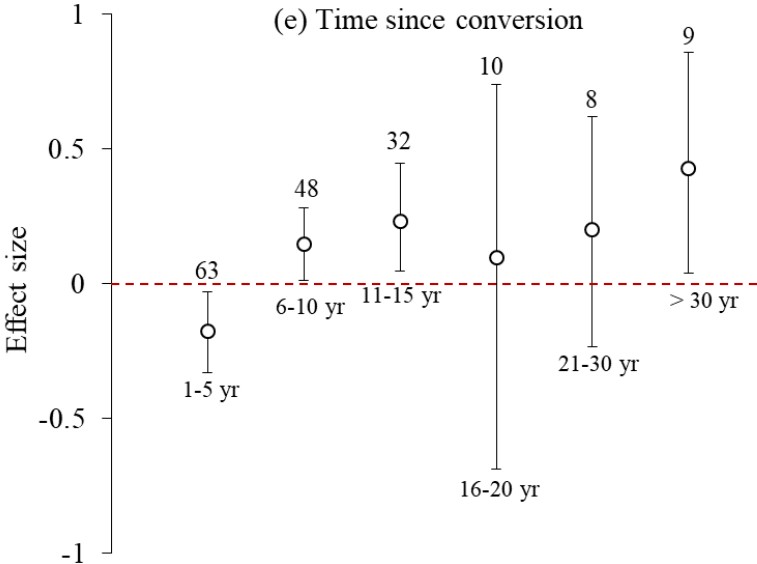





Figure 3

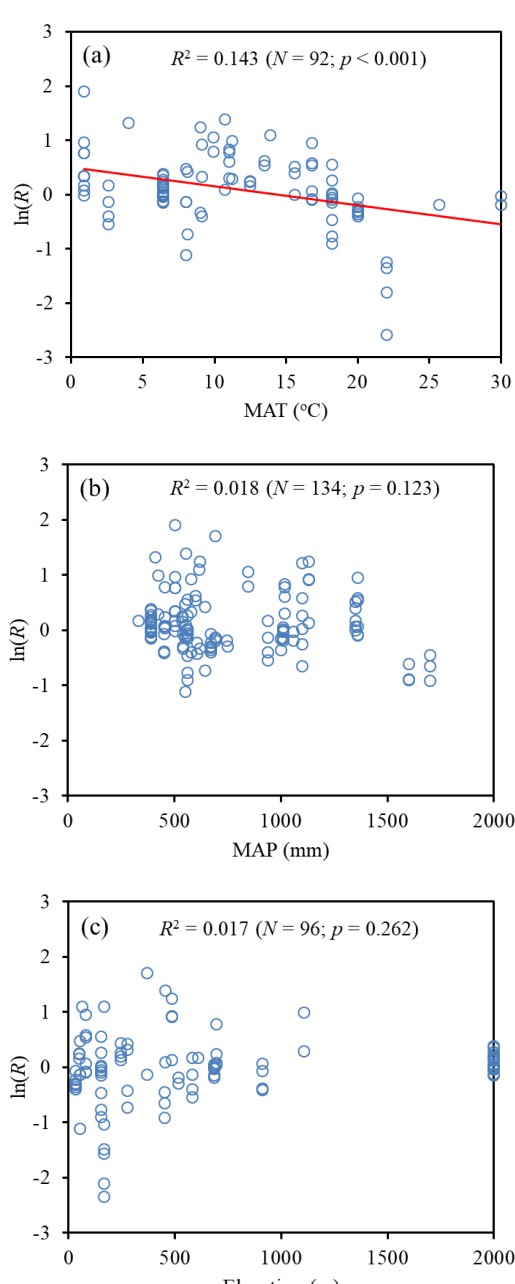