# Peer review of "Effects of environmental factors on the influence of tillage"

_SOIL, 2021_

## Referee Comment (RC1)

[Figure]

Figure 1

[Figure]

[Figure]

Figure 2

[Figure]

[Figure]

[Figure]

[Figure]

[Figure]

[Figure]

[Figure]

[Figure]

Figure 3

[Figure]

[referee-annotated manuscript omitted]

---

## Author Comment (AC1)

**Response to EC1**

1. This paper presents a meta-analysis of some papers that deal with the effects of a change from conventional agricultural tillage to some form of conservation tillage (reduced tillage or no tillage). From a literature search, 59 studies were selected following a screening process to include only those studies that provided key data on Ksat, the soil saturated hydraulic conductivity. This is taken by Liao et al. to be the same as steady-state infiltrability measured across the soil surface, though technically this is not the same thing at all. One refers to flow through a saturated porous medium, the other the imbibition of water from free water above the soil to pore water beneath the soil surface. In this case there are interface issues such as surface tension, surface crust and seal effects, the influence of litter, mulch, and other factors. I think that all of this could usefully be clarified in the present ms.

Answer: Thank you for your suggestion. In the revised manuscript, we have indicated that of the 65 studies, 7 did not provide $K_{sat}$ values, but steady-state infiltration rate values. The $K_{sat}$ refers to flow through a saturated porous medium, and the infiltration rate represents the imbibition of water from free water above the soil to pore water beneath the soil surface. In this case there are interface issues such as surface tension, surface crust and seal effects, the influence of litter, mulch, and other factors. Nevertheless, the steady-state infiltration rate was assumed to be the $K_{sat}$ by convention in this study (Yolcubal et al., 2004; Kirkham, 2014).

2. The paper seems to me to neglect some important issues that bear on the interpretation of the published studies. A serious issue for me is that there is no assessment of the quality of the data in the 59 studies. The authors tacitly accept all of the soil Ksat measurements as being valid and reliable measures of Ksat and suitable for their assessment of Ksat differences between forms of tillage. I do not think that this is a defensible position. It is widely-known, for instance, that the dimensions of the area of volume of soil tested influence the results of many Ksat (or steady infiltration rate) measurements. Thus, if ring or cylinder infiltrometers are used to

estimate Ksat from ponded conditions (whether single or double cylinder), the area of soil enclosed within the cylinders (expressed usually by the ring diameter) influences the result obtained. This makes intuitive sense, since a small cylinder might be underlain by a buried stone, so reducing the apparent Ksat, or by a large root macropore, so increasing the apparent Ksat. As the cylinder diameter is increased, the relative effect of such occurrences is reduced. Of course, the choice of an appropriate size of cylinder depends on the properties of the soil being tested. However, the point is that in this paper, Liao et al. simply accept all the results (not mentioning the cylinder diameter used) as valid and meaningful measurements. Unlike, for example, chemical properties, which with care can be measured precisely and unambiguously, the hydraulic properties of soils exhibit a complex dependency on the method and scale of measurement. Some authors have suggested that the measurements need to address something akin to the 'representative elementary volume' concept, adjusted to relate to the scale over which field conditions modulate Ksat.

I mention a few studies here that the authors might find helpful:

Fatehnia, M., Tawfiq, K., & Ye, M. (2016). Estimation of saturated hydraulic conductivity from double-ring infiltrometer measurements. European Journal of Soil Science, 67(2), 135-147. doi:https://doi.org/10.1111/ejss.12322

Please see Figure 6 in Fatehnia et al. (2016) for plots of infiltration rate vs ring or cylinder diameter.

Lai, J., & Ren, L. (2007). Assessing the Size Dependency of Measured Hydraulic Conductivity Using Double-Ring Infiltrometers and Numerical Simulation. Soil Science Society of America Journal, 71(6), 1667-1675. doi:https://doi.org/10.2136/sssaj2006.0227

This study concludes that inner ring diameters of > 80 cm are needed for reliable measurements.

Li, M., Liu, T., Duan, L., Luo, Y., Ma, L., Zhang, J., . . . Chen, Z. (2019). The Scale Effect of Double-Ring Infiltration and Soil Infiltration Zoning in a Semi-Arid Steppe. Water, 11(7), 1457. https://www.mdpi.com/2073-4441/11/7/1457

This is one of many additional studies of the infiltrometer scale effect.

Answer: Thank you for your suggestion. In the revised manuscript, for ring infiltrometer, the diameter of a single ring, or the diameter of the inner ring of a double ring, should be greater than 15 cm. This is because that Youngs (Journal of Soil Science, 38, 623–632, 1987) concluded that results were consistent from site to site when the ring size was at least 15 cm. Gregory et al. (Applied Turfgrass Science, 2, 1–7, 2005) also concluded that for a constant head test in sandy soil generally found in north and central Florida, a double-ring infiltrometer with 15-cm inner and 30-cm outer diameters would be suitable. In this case, the original selected literatures, such as Ouellet et al. (2008) and Abid and Lal (2009), have been deleted in the revised paper.

References:

1. Youngs, E.G. 1987. Estimating hydraulic conductivity values from ring infiltrometer measurements. Journal of Soil Science, 38, 623–632.

2. Gregory, J.H., Dukes, M.D., Miller, G.L. & Jones, P.H. 2005. Analysis of double-ring infiltration techniques and development of a simple automatic water delivery system. Applied Turfgrass Science, 2, 1–7.

3. Another concern that I have with this paper is the choice of test methods for the measurement of Ksat. The authors refer to the effects of drop impact on soil surfaces (e.g. lines 64-65, line 67) via the resulting sealing and crusting effects. Yet none of the measurement methods in their literature survey includes rainfall simulation on field plots, or the study of the response of 'natural plots' (those exposed to real rainfall). It is unclear why such effects should be excluded from analysis. It seems to me to be possible (perhaps probable) that by employing only static water, with no droplet impact, the methods used may well have over-estimated Ksat by excluding dynamic sealing and crusting effects. Likewise, intense rain can drive air into soil pores, thereby reducing infiltrability considerably. Field soils have some residual air content ('field saturation') which is not the same as the 'laboratory saturation' achieved by bottom-up wetting.

There are multiple published studies of conventional versus conservation tillage

that do indeed employ rainfall simulation. I have listed a couple of instances below.

A paper that reports rainfall simulation results, including hydraulic conductivity, in a study comparing traditional and conservation tillage is:

Packer, I., Hamilton, G., & Koen, T. (1992). Runoff, soil loss and soil physical property changes of light textured surface soils from long term tillage treatments. Soil Research, 30(5), 789-806. doi:https://doi.org/10.1071/SR9920789

It is not clear to me why papers such as this were not discovered in the literature survey by Liao et al. If they did not locate this paper, there may be many more that were also not located. It might take a wider choice of search terms than was adopted by Liao et al. to find relevant papers.

Some other examples of the application of rainfall simulation to the exploration of the effects of conservation tillage (neither is cited by Liao et al.) include:

Endale, D. M., Schomberg, H. H., Truman, C. C., Franklin, D. H., Tazisong, I. A., Jenkins, M. B., & Fisher, D. S. (2019). Runoff and nutrient losses from conventional and conservation tillage systems during fixed and variable rate rainfall simulation. Journal of Soil and Water Conservation, 74(6), 594. doi:10.2489/jswc.74.6.594

Salem, H. M., Ali, A. M., Wu, W., & Tu, Q. (2021). Initial effect of shifting from traditional to no-tillage on runoff retention and sediment reduction under rainfall simulation. Soil Research, -. doi:https://doi.org/10.1071/SR21082

Answer: Thank you for your suggestions. In the revised paper, we selected the studies that employ rainfall simulation methods (as shown in the figure below). It is found that for surface and subsurface soil $K_{sat}$, the mean effect sizes under CS conversion were 0.385 (95% CI: -0.033 to 0.766) and 0.314 (95% CI: 0.062 to 0.566) for rainfall simulator, respectively. The $K_{sat}$ measured by rainfall simulator tended to increase under CS practices. This is consistent with the findings of previous studies. For instance, Singh et al. (1994) observed that rainfall can reduce surface roughness, especially the first rains after tillage due to breakdown and sloughing of soil clods upon wetting during rainstorms. Therefore, Lampurlanés and Cantero-Martínez (2006) proposed that if a rainfall simulator had been used, greater infiltration rates would

probably have been found on NT, because residues play a role similar to that of surface roughness, i.e., increasing the time for infiltration to take place. However, Gupta et al. (1997) found the lower $K_{sat}$ values of soil in NT plots compared with those in CT plots, which was attributed to the fact that the NT practice allowed a consolidated layer to form. This was relatively impervious to the infiltrating water on the soil surface. The restricted downward movement of rain water produced lower $K_{sat}$ under NT. Therefore, more data are needed to test the effect of conversion to CS on $K_{sat}$ measured by rainfall simulator in the future.

[Figure]

Indeed, intense rain can drive air into soil pores, thereby reducing infiltrability considerably. However, $K_{sat}$ decreases under both CT and CS. Therefore, intense rain only affects the $K_{sat}$, but it is difficult to judge the effect of intense rain on the influence of tillage conversion on $K_{sat}$.

The study of Packer et al. (1992) has been used for meta-analysis, but the studies of Endale et al. (2019) and Salem et al. (2021) has not been applied since these two studies did not include saturated hydraulic conductivity data.

4. The acceptance of published data without evaluation of effects such as the scale of measurement, even when considering just one of the methods, viz., cylinder infiltrometry, and with the lack of reference to studies that employ rainfall simulation methods. The latter have many attendant issues, but at least may capture some of the effects of surface bombardment by water drops, and the development of air entrapment, seals, crusts, etc. In turn these issues suggest that the authors may need to cast their literature searching net somewhat wider than they appear to have done, as

there is a considerable relevant literature, and thorough searching is a cornerstone of thorough meta-analysis work.

Answer: Thank you for your suggestions. In the revised manuscript, for ring infiltrometer, the diameter of a single ring, or the diameter of the inner ring of a double ring, should be greater than 15 cm. This is because that Youngs (Journal of Soil Science, 38, 623–632, 1987) concluded that results were consistent from site to site when the ring size was at least 15 cm. Gregory et al. (Applied Turfgrass Science, 2, 1–7, 2005) also concluded that for a constant head test in sandy soil generally found in north and central Florida, a double-ring infiltrometer with 15-cm inner and 30-cm outer diameters would be suitable. In this case, the original selected literatures, such as Ouellet et al. (2008) and Abid and Lal (2009), have been deleted in the revised paper.

In addition, we also selected the studies that employ rainfall simulation methods (as shown in the figure below). It is found that for surface and subsurface soil $K_{sat}$, the mean effect sizes under CS conversion were 0.385 (95% CI: -0.033 to 0.766) and 0.314 (95% CI: 0.062 to 0.566) for rainfall simulator, respectively. The $K_{sat}$ measured by rainfall simulator tended to increase under CS practices. This is consistent with the findings of previous studies. For instance, Singh et al. (1994) observed that rainfall can reduce surface roughness, especially the first rains after tillage due to breakdown and sloughing of soil clods upon wetting during rainstorms. Therefore, Lampurlanés and Cantero-Martínez (2006) proposed that if a rainfall simulator had been used, greater infiltration rates would probably have been found on NT, because residues play a role similar to that of surface roughness, i.e., increasing the time for infiltration to take place. However, Gupta et al. (1997) found the lower $K_{sat}$ values of soil in NT plots compared with those in CT plots, which was attributed to the fact that the NT practice allowed a consolidated layer to form. This was relatively impervious to the infiltrating water on the soil surface. The restricted downward movement of rain water produced lower $K_{sat}$ under NT. Therefore, more data are needed to test the effect of conversion to CS on $K_{sat}$ measured by rainfall simulator in the future.

---

## Author Comment (AC2)

**Response to RC1**

1. I think that the paper is in general well written. However, I have some serious concerns about the election of factors that you relate with Ksat. Mean annual temperature does not affect Ksat, neither elevation does. The fact that you find a correlation between Ksat and MAT does not mean that mean annual temperature affects Ksat. It is probably a spurious correlation. You can say that Ksat and MAT were statistically correlated, but it does not mean that MAT controls Ksat. Otherwise, you should find and mention in the introduction several studies finding the same relation. You should select the factors to correlate with Ksat based on previous studies (in the introduction section there are not references relating Ksat to temperature nor to elevation for example). Probably, grouping the data in clusters could be helpful as well. There are several factors that were found to correlate with Ksat that were not considered as clay type, soil parent material, crop rotation, etc. This is a problem.

Answer: Thank you for your suggestions. In the revised manuscript, we have indicated that climatic and topographic factors mainly indirectly control $K_{sat}$ responses via other variables (e.g., soil moisture, biological processes and effective porosity) (Jarvis et al., 2013).

In the introduction section, we have specified that climatic and topographic factors were found to be related to $K_{sat}$. For instance, Jarvis et al. (2013) proposed that climatic factors can affect $K_{sat}$ through the effects of soil moisture on soil biota and plant growth and thus the abundance of root and faunal biopores; Yang et al. (2018) found that elevation and soil properties dominated $K_{sat}$ spatial distribution in the Loess Plateau of China.

In the revised paper, $K_{sat}$ of surface and subsurface soil layers was not analyzed together (Figs. 3 and 4). In addition, we also investigated the influence of cropping system management (single cropping and crop rotation) on the effect of tillage conversion on $K_{sat}$ (as shown in the figure below). It is found that cropping system management did not have a significant influence on the tillage effect on $K_{sat}$.

[Figure]

We did not investigate the influences of soil parent material and clay type on the effect of tillage conversion on $K_{sat}$, which is attributed to the fact that few literatures provided this information. In addition, we have considered the soil texture type, which is a function of the parent materials and clay type.

2. The title does not reflect the content of the manuscript. Authors should mention that it is a meta analysis on correlation between Ksat and environmental factors, including data obtained with different methodologies.

Answer: Sorry for this confusion. The title has been changed to "Effects of conversion from conventional tillage to conservation tillage on saturated soil hydraulic conductivity obtained with different methodologies: A global meta-analysis".

3. P4L71: Texture is not affected by tillage. Please rewrite.

Answer: Sorry for this confusion. "texture" has been replaced by "organic matter content".

4. P5L95: Even when a global analysis maybe is not yet available, there are several studies relating Ksat response to tillage and environmental conditions. Then you need to rewrite this sentence and cite these studies. For example:

https://doi.org/10.1016/j.geoderma.2013.04.015

https://doi.org/10.1016/j.still.2008.01.007

Answer: Thank you for your suggestions. In the revised paper, we have indicated

that previous studies have related the response of $K_{sat}$ to tillage and environmental conditions (Strudley et al., 2008; Bodner et al., 2013). However, there has not yet been a global synthetic analysis specifically focusing on how environmental conditions could affect the tillage effect on $K_{sat}$.

**References:**

Bodner, G., Scholl, P., Loiskandl, W., and Kaul, H.-P.: Environmental and

management influences on temporal variability of near saturated soil hydraulic

properties, Geoderma, 204-205, 120–129,

https://doi.org/10.1016/j.geoderma.2013.04.015, 2013.

Strudley, M. W., Green, T. R., and Ascough II, J. C.: Tillage effects on soil hydraulic

properties in space and time: State of the science, Soil Till. Res., 99, 4–48,

https://doi.org/10.1016/j.still.2008.01.007, 2008.

5. P6L106: Mean annual temperature and elevation are not factors influencing directly Ksat. Furthermore these factors were not justified in the introduction. Ksat of different soil layers can not be analyzed together. Furthermore, there are important factors as soil type, clay composition, crop rotation, that are very important in the response to different tillage systems.

Answer: Thank you for your suggestions. In the revised manuscript, we have indicated that climatic and topographic factors mainly indirectly control $K_{sat}$ responses via other variables (e.g., soil moisture, biological processes and effective porosity) (Jarvis et al., 2013).

In the introduction section, we have specified that climatic and topographic factors were found to be related to $K_{sat}$. For instance, Jarvis et al. (2013) proposed that climatic factors can affect $K_{sat}$ through the effects of soil moisture on soil biota and plant growth and thus the abundance of root and faunal biopores; Yang et al. (2018) found that elevation and soil properties dominated $K_{sat}$ spatial distribution in the Loess

Plateau of China.

In the revised paper, $K_{sat}$ of surface and subsurface soil layers was not analyzed together (Figs. 3 and 4). In addition, we also investigated the influence of cropping system management (single cropping and crop rotation) on the effect of tillage conversion on $K_{sat}$ (as shown in the figure below). It is found that cropping system management did not have a significant influence on the tillage effect on $K_{sat}$.

[Figure]

We did not investigate the influences of soil parent material and clay type on the effect of tillage conversion on $K_{sat}$, which is attributed to the fact that few literatures provided this information. In addition, we have considered the soil texture type, which is a function of the parent materials and clay type.

6. P8L149: This value is arbitrary of was calculated from the rest of the studies? Usually Ksat can show coefficient of variation higher than 0.5 (50 %) and sometimes higher than 100 %. Please justify the election of this value

Answer: Sorry for this confusion. This value is selected because it was suggested by a reviewer before. However, we don't really agree with him/her. Because the coefficient of variation here does not refer to the spatial variation of saturated hydraulic conductivity, but is calculated for repeated samples. Therefore, according to most previous studies, if the SD value is not given in the original text, it is generally 0.1 times the mean value by default.

For example:

Li, Y., Li, Z., Cui, S., Jagadamma, S., and Zhang, Q.: Residue retention and minimum tillage improve physical environment of the soil in croplands: A global

meta-analysis, Soil Till. Res., 194, 104292, https://doi.org/10.1016/j.still.2019.06.009, 2019.

**2.2. Data analysis**

Data were analyzed by calculating the natural logarithm of the response ratio ($\ln RR$) for each soil property index to compare treatment means (-NT, RT, NTS, RTS, or CTS) with the control means (-CT) (Osenberg et al. (1999); Li et al. (2018). The variance ($v$) of the $\ln RR$ was calculated using the equation, $v = S_t^2/n_t \times X_t^2 + S_c^2/n_c \times X_c^2$, where $S_t$ and $S_c$ represent the standard deviations of the treatment and control groups, respectively; and where $n_t$ and $n_c$ represent the number of replicates for the treatment and the control groups, respectively. For studies that did not report SD or SE, SD was estimated as 0.1 times the mean (Luo et al., 2006). To derive the overall response effect of the treatment group compared to the control group, the weighted response ratio ($RR_{++}$, also defined as effect size) between the treatment and control groups was calculated according to Hedges et al. (1999); Luo et al. (2006), as described in Equation (1):

$$RR_{++} = \frac{\sum_{i=1}^{m} \sum_{j=1}^{k} w_{ij} RR_{ij}}{\sum_{i=1}^{m} \sum_{j=1}^{k} w_{ij}} \qquad (1)$$

In addition, we also studied the influence of SD value (0.1, 0.2, 0.4 and 1 times the mean) on the meta-analysis results, and found that it is not significant.

Therefore, in the revised paper, we still select 0.1 times the mean as the standard deviation.

7. P11L218: Please see my observation in the figure. The linear regression does not seem to be a good election for the data.

Answer: Indeed, the accuracy of quadratic polynomial fitting is higher than that of linear regression. In the revised manuscript, we have indicated that the relationships between the $\ln(R)$ of $K_{sat}$ and MAT, MAP, and elevation can be well fitted by quadratic polynomials, with the $R^2$ values ranging between 0.064 and 0.585 (Fig. 4).

8. P12L234-235: In the Discussion section it is important to mention that since studies comparing CS vs CT effects on Ksat using different methods are from different places, maybe there are other reasons that explain the differences found. For example if studies from Argentinean pampas region do not include some methodologies, maybe in those soils the results are not only affected by the

methodology, temperature and precipitation, but also by the clay type or other factors. Some cold weather soils present freezing-thawing processes that are important for pore generation. And so on. Please mention all these possible reasons for the results.

Answer: Thank you for your suggestion. In the revised manuscript, we have indicated that since studies comparing tillage conversion effects on $K_{sat}$ using different methodologies are from different places, maybe there are other reasons that explain the differences found. For example, the study of Lozano et al. (2016) from Argentinean pampas region did not include ring infiltrometer, hood infiltrometer and rainfall simulator, maybe in those soils the results are not only affected by the measurement technique, MAT and MAP, but also by the clay type or other factors. Some cold weather soils present freezing-thawing processes that are important for pore generation.

9. Conclusions: These conclusions are more results than real conclusions. It would be better if you write an explicit hypothesis (at the end of the introduction section). Then the conclusions will be an answer to the hypothesis.

Answer: Thank you for your suggestion. At the end of the introduction section, we specifically hypothesized that conversion to CS can increase the soil $K_{sat}$ measured by ring infiltrometer and rainfall simulator. In the conclusions section, we have indicated that the effect of tillage conversion on $K_{sat}$ was related to experimental conditions, especially the measurement technique, conversion period and climatic and topographic factors. The increase of $K_{sat}$ measured by single- or double-ring infiltrometer and rainfall simulator was substantially larger than the other techniques.

10. P15L315: This reference is not cited in the manuscript. Remove
Answer: Sorry for this confusion. This reference has been removed in the revised manuscript.

11. P16L335: This reference is not cited in the manuscript. Remove
Answer: Sorry for this confusion. This reference has been removed in the revised

manuscript.

12. P18L366: This reference is not cited in the manuscript. Remove

Answer: Sorry for this confusion. This reference has been removed in the revised manuscript.

13. P20L416: This reference is not cited in the manuscript. Remove

Answer: Sorry for this confusion. This reference has been removed in the revised manuscript.

14. Figure 3: The relationship between ln R and MAT does not seem to be linear. The residuals of the regression should be independent and present homoscedasticity. Maybe another function adjust better.

Answer: Thank you very much for you suggestion. Indeed, the accuracy of quadratic polynomial fitting is higher than that of linear regression. In the revised manuscript, we have indicated that the relationships between the $\ln(R)$ of $K_{sat}$ and MAT, MAP, and elevation can be well fitted by quadratic polynomials, with the $R^2$ values ranging between 0.064 and 0.585 (Fig. 4).

---

## Author Comment (AC3)

**Response to RC2**

1. I consider that the topic is within the scope of the journal and has international relevance. The manuscript is in general well-structured and written. However, I agree with the concerns presented by the editor and the reviewer 1.

Answer: Thank you for your suggestions. We have revised the manuscript according to the concerns presented by the editor and the reviewer 1.

2. Page 4, Line 70. Is this sentence implying that tillage affects soil texture?

Answer: Sorry for this confusion. In fact, texture is not affected by tillage. Therefore, "texture" has been changed to "organic matter content".

3. Page 10, Line 203-205. Suggestion: add the treatments that are being compared, to the sentence.

Answer: Thank you for your suggestion. In the revised manuscript, we have indicated that for surface soil $K_{sat}$, the mean effect sizes under CS conversion were 0.039 (95% CI: -0.543 to 0.661), -0.002 (95% CI: -0.086 to 0.075), 0.307 (95% CI: 0.079 to 0.561), -0.130 (95% CI: -0.441 to 0.124), 0.045 (95% CI: -0.186 to 0.268) and 0.385 (95% CI: -0.033 to 0.766) for hood infiltrometer, tension disc infiltrometer, ring infiltrometer, constant/falling head, Guelph permeameter and rainfall simulator, respectively (Fig. 3a). However, the mean effect sizes of subsurface $K_{sat}$ under CS conversion were 0.234 (95% CI: -0.364 to 0.800), -0.131 (95% CI: -0.314 to 0.123), 0.036 (95% CI: -0.188 to 0.249), 0.212 (95% CI: -0.026 to 0.466), and 0.314 (95% CI: 0.062 to 0.566) for tension disc infiltrometer, ring infiltrometer, constant/falling head, Guelph permeameter and rainfall simulator, respectively (Fig. 3b).

4. Page 11, Line 208. It is suggested to replace the wording "under CS" for "under CS conversion".

Answer: Thank you for your suggestion. As the content of the paper has been greatly revised according to the comments from the editor and reviewers, this statement has been deleted in the revised paper.

5. Line 210. Consider an alternative for this wording: reverse response

Answer: Sorry for this confusion. As the content of the paper has been greatly revised according to the comments from the editor and reviewers, this statement has been deleted in the revised paper.

6. Line 261. Eliminate the word "which" or revise the sentence for alternatives.

Answer: Sorry for this error. In the revised manuscript, we have indicated that soil compaction under CS can lead to a reduction in macroporosity and an increase in bulk density and microporosity.

---

## Author Response (AR2)

Dear Editor,

Thank you very much for your and reviewers' efforts on our paper submitted to the "Soil" (Manuscript ID soil-2021-118). We have checked the manuscript and revised it according to the comments carefully. The revision has been highlighted in the document by using colored text. We submit here the revised manuscript as well as an itemized response to reviewers' comments.

Sincerely yours,

Dr. Kaihua Liao

**Response to EC1**

1. I think that these have resulted in worthwhile improvements. However, the revisions made are relatively minor. For instance, you have not searched for additional papers employing the rainfall simulation method, but have only considered the three example papers (there are likely to be many more) that I suggested. A meta-analysis should I think be based on the largest, and therefore most representative, set of data that can be assembled. Therefore, I would like to ask you to undertake additional literature searching to more fully include the results of rainfall simulation studies.

Answer: Thank you very much for your suggestion. According to your comment, we have greatly revised the paper. Additional papers employing the rainfall simulation method have been included (Table 1). We applied a global meta-analysis method to synthesize a total of 37 paired observations for soil $K_{sat}$ measured by rainfall simulator.

[Figure]

It is found that when the $K_{sat}$ was measured by rainfall simulator, conversion to CS significantly ($p < 0.05$) increased the surface and subsurface soil $K_{sat}$ by 41.7% and 36.9%, respectively. This is consistent with the findings of previous studies. For instance, Singh et al. (1994) observed that rainfall can reduce surface roughness, especially the first rains after tillage due to breakdown and sloughing of soil clods upon wetting during rainstorms. Lampurlanés and Cantero-Martínez (2006) proposed that if a rainfall simulator had been used, greater infiltration rates would probably have been found on NT, because residues play a role similar to that of surface roughness, i.e., increasing the time for infiltration to take place. However, Gupta et al. (1997) found the lower $K_{sat}$ values of soil measured by rainfall simulator in NT plots compared with those in CT plots, which was attributed to the fact that the NT practice allowed a consolidated layer to form. This was relatively impervious to the infiltrating water on the soil surface. The restricted downward movement of rain water produced lower $K_{sat}$ under NT. Therefore, more data are needed to test the effect of conversion to CS on $K_{sat}$ measured by rainfall simulator in the future (P14L258-271).

**Table 1** Rainfall simulator technique used for measuring saturated soil hydraulic conductivity.

| Reference | $N^a$ | Measurement technique[b] | Measurements | | OMC[e] (%) | Time interval from tillage conversion to measurement (yr) | General descriptions |
|---|---|---|---|---|---|---|---|
| | | | $IR^c$ | $K_{sat}{}^d$ | | | |
| Baumhardt et al. (2012) | 4 | Rainfall simulator | √ | | | 3 | We applied reverse-osmosis water in lieu of rainwater because of its dispersive characteristics at a rate of 78 mm h$^{-1}$ with a rotating-disk rain simulator that produced impact energy of 22 J mm$^{-1}$ m$^{-2}$ or 80% of natural rain. |
| Curtis and Claassen (2009) | 4 | Rainfall simulator | | √ | 0.03~7.00 | 2 (2003~2004) | |
| De Almeida et al. (2018) | 1 | Rainfall simulator | √ | | 2.55~7.28 | 0.5 (November 2013~May 2014) | We use a portable rainfall simulator calibrated with a constant rain intensity of 60 ± 1.715 mm h$^{-1}$, mean drop diameter of 2.0 mm and pressure of 32 kPa. |
| Gómez et al. (1999) | 2 | Rainfall simulator | √ | √ | 1.00~2.50 | 16 (1982~1997) | The $K_{sat}$ was calculated at four water tensions; -15, -10, -5 and 0 cm of H$_2$O. |
| Gupta et al. (1997) | 1 | Rainfall simulator | | √ | 0.00~3.40 | 2 (1992~1993) | The rainfall was applied by the portable rainfall simulator. The infiltration data was analyzed for the saturated hydraulic conductivity determinations. |
| Jakab et al. (2017) | 15 | Rainfall simulator | √ | | | 15 | Rainfall simulation was carried out three times in 2016. |
| Langhans et al. (2011) | 1 | Rainfall simulator | √ | | 1.09~1.38 | | Prior to each experiment, 45 mm h$^{-1}$ of rainfall from 3.25 m height from a nozzle-type simulator was applied to the plot until steady state runoff occurred. |

| Reference | | Method | | | | | Description |
|---|---|---|---|---|---|---|---|
| Maulé and Reed (1993) | 3 | Rainfall simulator | | √ | | 13 | The rainfall occurred over a 1.6 m x 2.3 m area on the ground, but infiltration was determined from a 1.5 m² area directly under the simulator. |
| McGarry et al. (2000) | 1 | Rainfall simulator | | √ | | 8 | Water infiltration parameters were measured with an oscillating nozzle rainfall simulator. |
| Ndiaye et al. (2005) | 6 | Rainfall simulator | | √ | < 0.5 | | The rainfall simulator was a 4 m high tower equipped with a Laechler nozzle (# 461.008.30) mounted at 3.86 m above the soil surface. |
| Nyamadzawo et al. (2007) | 5 | Rainfall simulator | √ | | ~1.72 | 2 (2001~2002) | Rainfall simulations were done at the centre of the plots at a rainfall intensity of 35 mm h⁻¹ on 1 m² experimental plots surrounded by a 50 cm buffer zone. |
| Park et al. (1992) | 4 | Rainfall simulator | | √ | | 7 (1981~1987) | Rainfall simulation studies were conducted on each tillage treatment using a rotating disk simulator. |
| Potter et al. (1995) | 3 | Rainfall simulator | | √ | | 8 | A rainfall simulator was used to determine water infiltration characteristics of a Houston Black clay. |
| Ramos et al. (2019) | 6 | Rainfall simulator | √ | | 0.57~4.31 | 3 | The susceptibility to sealing of each soil and the steady infiltration rates were evaluated in the laboratory subjecting the soils to rainfall simulation applied at an intensity of 25 mm h⁻¹. |
| So et al. (2009) | 1 | Rainfall simulator | | √ | 2.88 | 14 | A rainfall simulator with an intensity of 80 mm h⁻¹ was used to determine the infiltration characteristics of the bare soil using plots of 2 m x 1.5 m |
| TerAvest et al. (2015) | 2 | Rainfall simulator | √ | | 0.29~2.79 | 3 | Rainfall simulations were conducted 6–7 weeks after planting, between crop rows, when soils were at or near field capacity. |
| Xin et al. (2005) | 4 | Rainfall simulator | | √ | | 4 (2001~2004) | The swing sprinkler rainfall simulator produced by |

Queensland Department of primary industries is adopted. The sprinkler model of rainfall simulator (RFs) is Veeject 80100, with a total of 3 nozzles.

[a] Number of paired observation for $K_{sat}$; [b] The numbers in parentheses indicate the diameter (cm) of the device. For double ring method, the diameters of inner and outer rings are provided; [c] Infiltration rate; [d] Saturated hydraulic conductivity; [e] Organic matter content. If the literature only provided the organic carbon content, the organic matter content was estimated using the 1.724 conversion factor.

2. A second issue is the scale of measurement, which I raised initially. The various methods examine the properties of different areas of the soil surface: from plots to small core samples. How can this be addressed, so as to understand the potential influence of the scale of measurement? The issue of the diameter of cylinder infiltrometers is just one instance of this effect. You have chosen to rely on a rather old paper or two, and ignore the wider body of evidence that suggests that scale effects are important, and that large diameter rings must be used. What I would like to ask you to do is to actually consider this literature, and acknowledge more adequately that 15 cm diameter is regarded as far too small by many authors. A key issue is to acknowledge that there is a scale effect, and that this may exert an influence on the results.

Answer: Thank you very much for your suggestion. Indeed, scale effects are very important. A total of 12 articles were retrieved from the Web of Science database, using an inner ring of more than 80 cm. However, most of the articles only discussed the influence of the size of double ring on the measurement results of $K_{sat}$, while only one article investigated the influence of tillage conversion on $K_{sat}$.

In the revised manuscript, we have indicated that for ring infiltrometer, the diameter of a single ring, or the diameter of the inner ring of a double ring, should be greater than 50 cm in this study (Table 2), although inner and outer ring diameters of about 30 and 60 cm have been widely applied to measure the soil infiltration process (e.g., Ronayne et al., 2012; Zhang et al., 2017). A recent study (Li et al., 2019b) have demonstrated that the ring infiltrometer with an inner diameter of 40 cm is not enough to completely overcome the scale effect (P8L139-145).

After using the inner ring with longer diameter, it is found that the analysis results are different from those before. In the revised paper, we observed that conversion to CS had no significant effect on the soil $K_{sat}$ measured by ring infiltrometer.

Finally, the Figure 1 has also been revised since the selected literature has changed.

**Table 2** The ring infiltrometer technique used for measuring saturated soil hydraulic conductivity.

| Reference | $N$[a] | Measurement technique[b] | Measurements | | OMC[e] (%) | Time interval from tillage conversion to measurement (yr) | General descriptions |
|---|---|---|---|---|---|---|---|
| | | | $IR$[c] | $K_{sat}$[d] | | | |
| Kennedy and Schillinger (2006) | 9 | Single ring (76) | √ | | 1.28~2.88 | 20 | A 76-cm-diameter single-ring infiltrometer was pushed into the soil to a depth of 5 cm. |
| Khorami et al. (2018) | 1 | Double ring (50~70) | √ | | 1.48~1.98 | 3 (2014~2016) | The double-ring infiltrometer test procedure involved inserting two rings into the 5-cm depth of the soil. |
| Merwin et al. (1994) | 3 | Single ring (70) | | √ | 5.30 | 8 | A 70-cm-diameter single-ring infiltrometer was pushed into the soil to a depth of 8 cm. |
| Sartori et al. (2022) | 1 | Double ring (an inner ring of 60 cm in diameter) | | √ | 1.41 | 3 | An inner ring of 60 cm in diameter was used to measure both the row and inter-row areas in the tillage radish plots. |
| Smith et al. (2007) | 1 | Double ring (50~90) | | √ | | 1 | A double-ring infiltrometer with two concentric metal rings were co-located with tensiometers placed at depths of 15, 30 and 60 cm in the soil profile located just outside the inner ring. |
| Starr and Glotfelty (1990) | 4 | Double ring (75~150) | √ | | 2.40~3.00 | 13 (1974~1986) | Each microplot was instrumented with a double-ring infiltrometer that was pushed into the soil to a depth of about l0 cm. |
| Steenhuis et al. (1990) | 1 | Single ring (70) | | √ | | 3 | A 70-cm-diameter single-ring infiltrometer was pushed into the soil to a depth of 5 cm. |
| Stone and Schlegel (2010) | 2 | Double ring (92~124) | √ | | 1.90 | 12 (1989~2000) | Rings were positioned to avoid vehicle traffic paths, driven 13 cm deep, and filled twice with water. At |

sunup 2 d later, water was added to the infiltrometers, and ponding was maintained at a depth of ~3 to 10 cm.

[a] Number of paired observation for $K_{sat}$; [b] The numbers in parentheses indicate the diameter (cm) of the device. For double ring method, the diameters of inner and outer rings are provided; [c] Infiltration rate; [d] Saturated hydraulic conductivity; [e] Organic matter content. If the literature only provided the organic carbon content, the organic matter content was estimated using the 1.724 conversion factor.

[Figure]

Before revision                After revision

**Figure 1:** The geographical coverage of the studies used in the meta-analysis.